# Identifying proteomic risk factors for cancer using prospective and exome analyses of 1463 circulating proteins and risk of 19 cancers in the UK Biobank

Keren Papier [1,4] ✉, Joshua R. Atkins [1,4], Tammy Y. N. Tong [1], Kezia Gaitskell [1], Trishna Desai[1], Chibuzor F. Ogamba [1], Mahboubeh Parsaeian[1], Gillian K. Reeves[1], Ian G. Mills [2,3], Tim J. Key [1], Karl Smith-Byrne [1,4] & Ruth C. Travis[1,4]

The availability of protein measurements and whole exome sequence data in the UK Biobank enables investigation of potential observational and genetic protein-cancer risk associations. We investigated associations of 1463 plasma proteins with incidence of 19 cancers and 9 cancer subsites in UK Biobank participants (average 12 years follow-up). Emerging protein-cancer associations were further explored using two genetic approaches, *cis*-pQTL and exome-wide protein genetic scores (exGS). We identify 618 protein-cancer associations, of which 107 persist for cases diagnosed more than seven years after blood draw, 29 of 618 were associated in genetic analyses, and four had support from long time-to-diagnosis (> 7 years) and both *cis*-pQTL and exGS analyses: CD74 and TNFRSF1B with NHL, ADAM8 with leukemia, and SFTPA2 with lung cancer. We present multiple blood protein-cancer risk associations, including many detectable more than seven years before cancer diagnosis and that had concordant evidence from genetic analyses, suggesting a possible role in cancer development.

Proteins are integral to most biological processes including many that lead to carcinogenesis, such as tissue growth and proliferation. Previous prospective studies of individual or small panels of blood proteins have identified aetiological cancer proteins, such as insulin-like growth factor-I, which is a causal risk factor for breast, colorectal, and prostate cancers, and microseminoprotein-beta, which is associated with lower prostate cancer risk[1–3]. Other cancer biomarkers identified include protein markers for early detection, progression, recurrence and prognosis, for example, CA-125, CEACAM5, CA19-9 and prostate-specific antigen[4–7]. However, new multiplex proteomics methods allow for the simultaneous measurement of thousands of proteins, many of

which have not previously been assessed for their associations with risk across multiple cancer sites.

Identifying aetiological markers of cancer risk using prospective data alone can be challenging due to the potential for confounding and other epidemiological biases. However, the abundance of many proteins in the circulation can be partially explained by inherited genetic variation; these genetic predictors of protein levels can be used to generate complementary evidence, with orthogonal biases, on protein-cancer associations[8–10]. Many of these genetic variants lie in a protein's cognate gene (known as *cis* protein quantitative trait loci [*cis*-pQTL]) and likely influence biological processes directly and can be

[1]Cancer Epidemiology Unit, Nuffield Department of Population Health, University of Oxford, Oxford, UK. [2]Nuffield Department of Surgical Sciences, University of Oxford, Oxford, UK. [3]Patrick G Johnston Centre for Cancer Research, Queen's University Belfast, Belfast, UK. [4]These authors contributed equally: Keren Papier, Joshua R. Atkins, Karl Smith-Byrne, Ruth C. Travis. ✉e-mail: keren.papier@ndph.ox.ac.uk

highly robust and specific predictors of protein concentrations[11–13]. Such genetic analyses complement traditional prospective epidemiology, and the combination of observational and genetic approaches can improve our ability to identify proteins most likely to have a causal role in cancer development and progression[14].

Here, we use an integrated multi-omics approach combining prospective cohort and exome-variant study designs to identify proteins with a role in cancer aetiology: we describe the association of 1463 protein biomarkers quantified using the Olink platform with the risk of 19 common cancers and 9 cancer subsites in 44,645 UK Biobank participants, overall and by time to diagnosis. We further assess the identified protein-cancer associations as aetiological risk factors using exome *cis*-pQTL variant and exome-wide genetic score analyses (exGS).

## Results

### Observational analyses

Our prospective analyses included 4921 incident malignant cancer cases with a mean follow-up of 12 years (SD 2.7). The median age at any cancer diagnosis was 66.9 years (Interquartile range (IQR) 9.9) [youngest median diagnosis was for breast cancer in women (median 64.5, IQR 12.5) and oldest for squamous cell carcinoma of the lung in women (median 71.8, IQR 9.9)]. Supplementary Data 1 shows the median ages at diagnosis for all cancer subsites.

Baseline characteristics of the analysis sample overall, by sex and in those who developed a malignant cancer over follow-up are shown in Table 1 and Supplementary Data 2. Compared with the total analysis sample, participants who developed cancer were on average older and

a higher proportion of them were former or current smokers, moderate to high alcohol consumers, and had a family history of various cancers; among the women, they reported having fewer children, were younger at menarche, and a higher proportion of them were postmenopausal, had used hormone replacement therapy, and had never used the oral contraceptive pill.

From the 1463 proteins included in our analyses, we identified an association for 371 proteins with a risk of at least one cancer after correction for multiple testing, which amounted to 618 protein-cancer associations (Fig. 1, Fig. 2 & Supplementary Data 3). Almost half of these associations (304) were for proteins enriched (greater than 10% of total body expression) for mRNA expression in either the tissue or candidate cell of origin for cancer indicated in our analyses. For 83 of the protein-cancer associations, the proteins whose cognate genes were majority expressed (i.e., > 50%) in either the tissue or candidate cell of origin. Many of these associations were for proteins that were associated with the risk of haematological cancers with high mRNA expression in either B-cells or T-cells. However, we also identified proteins that were both associated with risk for cancer and either had enriched or majority mRNA expression in the liver, lung, colorectum, kidneys, brain, stomach, oesophagus, and endometrium (Supplementary Fig. 6).

More than half of our ENT-significant protein-cancer associations (320) were for haematological malignancies (non-Hodgkin overall (NHL) [124], diffuse large B-cell non-Hodgkin (DLBCL) [50], leukaemia [87], and multiple myeloma [59]). These included the associations of TNFRSF13B and SLAMF7 with risk of multiple myeloma [HR (95%CI): 2.09 (1.96–2.24) and 3.07 (2.73–3.46), respectively], PDCD1 and

**Table 1 | Baseline characteristics of the UK Biobank analysis cohort, overall, by sex, and in those who developed any malignant cancer**

| Characteristics | All (N = 44,645) | Women (n = 23,274) | Men (n = 21,371) | Developed a malignant cancer (n = 4921) |
|---|---|---|---|---|
| Sociodemographic | | | | |
| Age (years) | 57.0 (8.3) | 57.0 (8.1) | 57.1 (8.4) | 60.6 (7.0) |
| Townsend deprivation, *n* (%) | | | | |
| Most deprived | 9416 (21.1%) | 4797 (20.6%) | 4619 (21.6%) | 1066 (21.7%) |
| Lifestyle | | | | |
| Physical activity level, *n* (%) | | | | |
| High ≥50 METs | 7927 (17.8%) | 3644 (15.7%) | 4283 (20.0%) | 852 (17.3%) |
| Smoking, *n* (%) | | | | |
| Never | 24,481 (54.8%) | 13,980 (60.1%) | 10,501 (49.1%) | 2220 (45.1%) |
| Current ≥15 cigarettes/day | 1818 (4.1%) | 752 (3.2%) | 1066 (5.0%) | 315 (6.4%) |
| Alcohol intake, *n* (%) | | | | |
| non-drinkers | 3586 (8.0%) | 2240 (9.6%) | 1346 (6.3%) | 359 (7.3%) |
| ≥20 g/day | 12,563 (28.1%) | 3230 (13.9%) | 9333 (43.7%) | 1578 (32.1%) |
| Anthropometric | | | | |
| Standing height in cm | 168.7 (9.3) | 162.4 (6.4) | 175.6 (6.9) | 169.6 (9.1) |
| Body mass index (kg/m²) | 27.3 (4.6) | 27.0 (5.1) | 27.6 (4.0) | 27.6 (4.6) |
| Women's Health | | | | |
| Parity in women, *n* (%) | | | | |
| Nulliparous | – | 4257 (9.5) | – | 416 (19.1%) |
| Age at first menarche in women, *n* (%) | | | | |
| <12 years | – | 4467 (19.2%) | – | 441 (20.3%) |
| Menopausal status in women, n (%) | | | | |
| Postmenopausal | – | 16,580 (71.2%) | – | 1760 (80.9%) |
| Hormone replacement therapy use in women, *n* (%) | | | | |
| Never | – | 15,036 (64.6%) | – | 1256 (57.7%) |
| Oral contraceptive pill use in women, *n* (%) | | | | |
| Never | – | 4683 (20.1) | – | 490 (22.5) |

Values are presented as mean (standard deviation) unless otherwise specified.

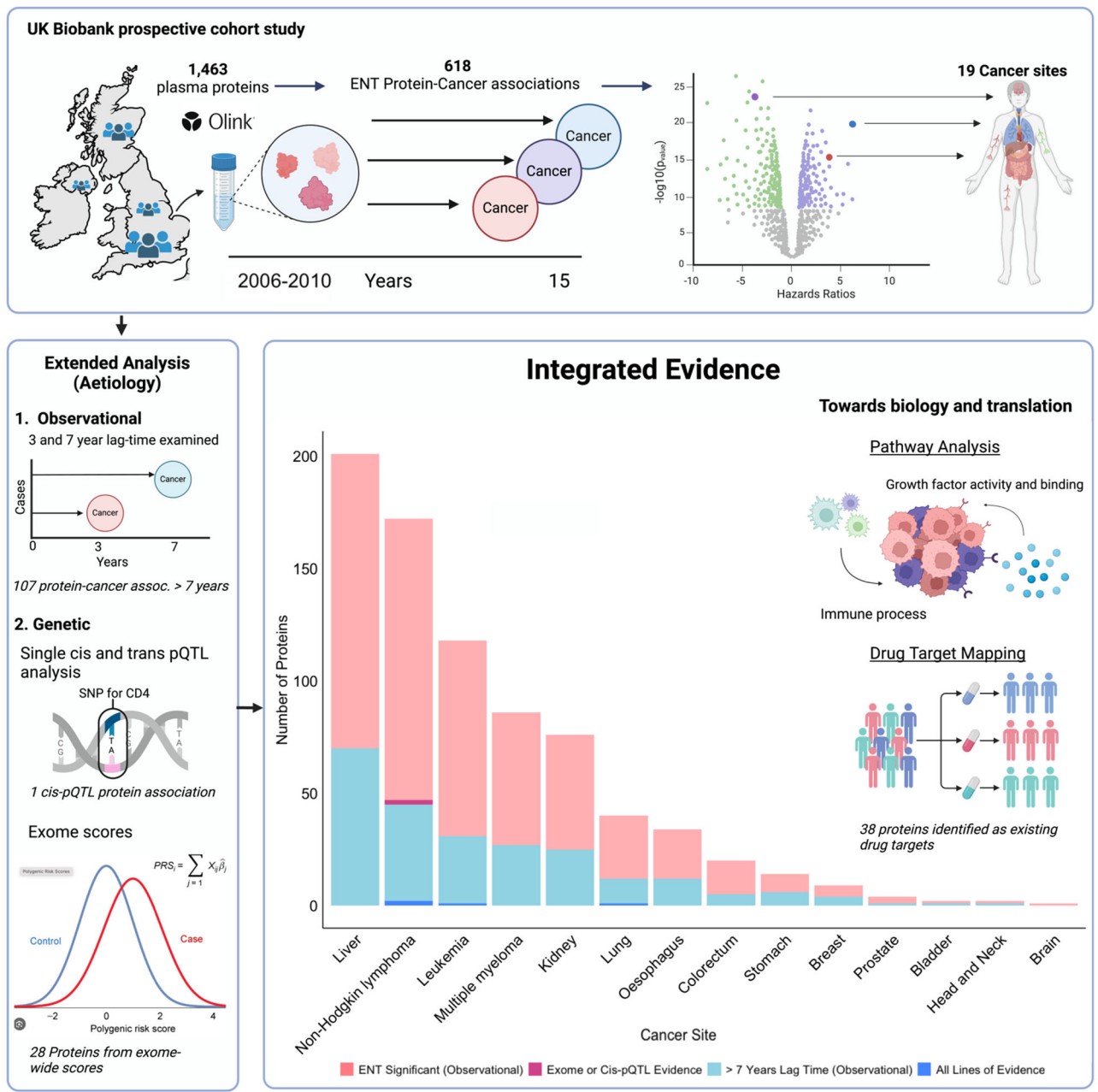

**Fig. 1 | Summary of study design, observational and genetic protein-cancer associations, and pathway analyses and drug target mapping.** cis-pQTL - cis protein quantitative trait loci, PRS – polygenic risk score, SNP – single nucleotide polymorphism, ENT – effective number of tests. Source data are provided as a Source Data file.

TNFRSF9 with risk of NHL [1.99 (1.87–2.11) and 1.98 (1.85–2.11), respectively], and FCER2 and FCRL2 with risk of leukaemia [2.12 (1.98–2.29) and 2.10 (1.95–2.26), respectively].

We also observed associations between 131 proteins and risk of liver cancer that included IGFBP7 and IGFBP3 [1.65 (1.48–1.84) and 0.46 (0.39–0.54), respectively], and 51 proteins and risk of kidney cancer, such as HAVCR1 and ESM1 [2.88 (2.55–3.24) and 1.84 (1.55–2.19)]. We identified 28 proteins associated with the risk of lung cancer overall and/or at least one histological subtype that included WFDC2 and CEACAM5 [1.52 (1.39–1.67) and 1.44 (1.33–1.56)]. Although most protein-cancer associations (log odds) did not differ greatly between minimally and fully adjusted models, some proteins associated with the risk of lung cancer after ENT correction were attenuated by more than 50% compared with minimally adjusted models, which may imply a potential risk for residual confounding stemming from measurement error in smoking behaviours (Supplementary Fig. 2).

Twenty-two proteins were associated with the risk of oesophageal cancer and/or oesophageal adenocarcinoma, including REG4 and ST6GAL1 [2.02 (1.66–2.45) and 1.83 (1.53–2.19)]. We identified 15 proteins associated with colorectal, colon, and/or rectal cancer, such as AREG and GDF15 [1.30 (1.19–1.42) and 1.32 (1.20–1.45)]. Five proteins were associated with the risk of stomach cancer including ANXA10 and TFF1 [1.75 (1.51–2.02) and 1.90 (1.58–2.28)]. We found five proteins associated with the risk of breast cancer, such as STC2 and CRLF1 [1.33 (1.23–1.44) and 1.31 (1.21–1.42)]. Three proteins were associated with risk of prostate cancer: GP2, TSPAN1, and FLT3LG [1.29 (1.21–1.36), 1.14 (1.09–1.18), and 0.87 (0.82–0.92)] and three were associated with endometrial cancer: CHRDL2, KLK4, and WFIKKN1 [1.42 (1.21–1.65), 1.41 (1.20–1.65), and 1.42 (1.20–1.68)]. Two proteins were associated with the risk of ovarian cancer, DKK4 and WFDC2 [1.46 (1.28–1.70), 1.57 (1.26–1.96)]. We identified one protein for each of bladder [WAS, 0.54 (0.39–0.73)], brain [GFAP, 1.55 (1.31–1.86)], and head and neck cancers

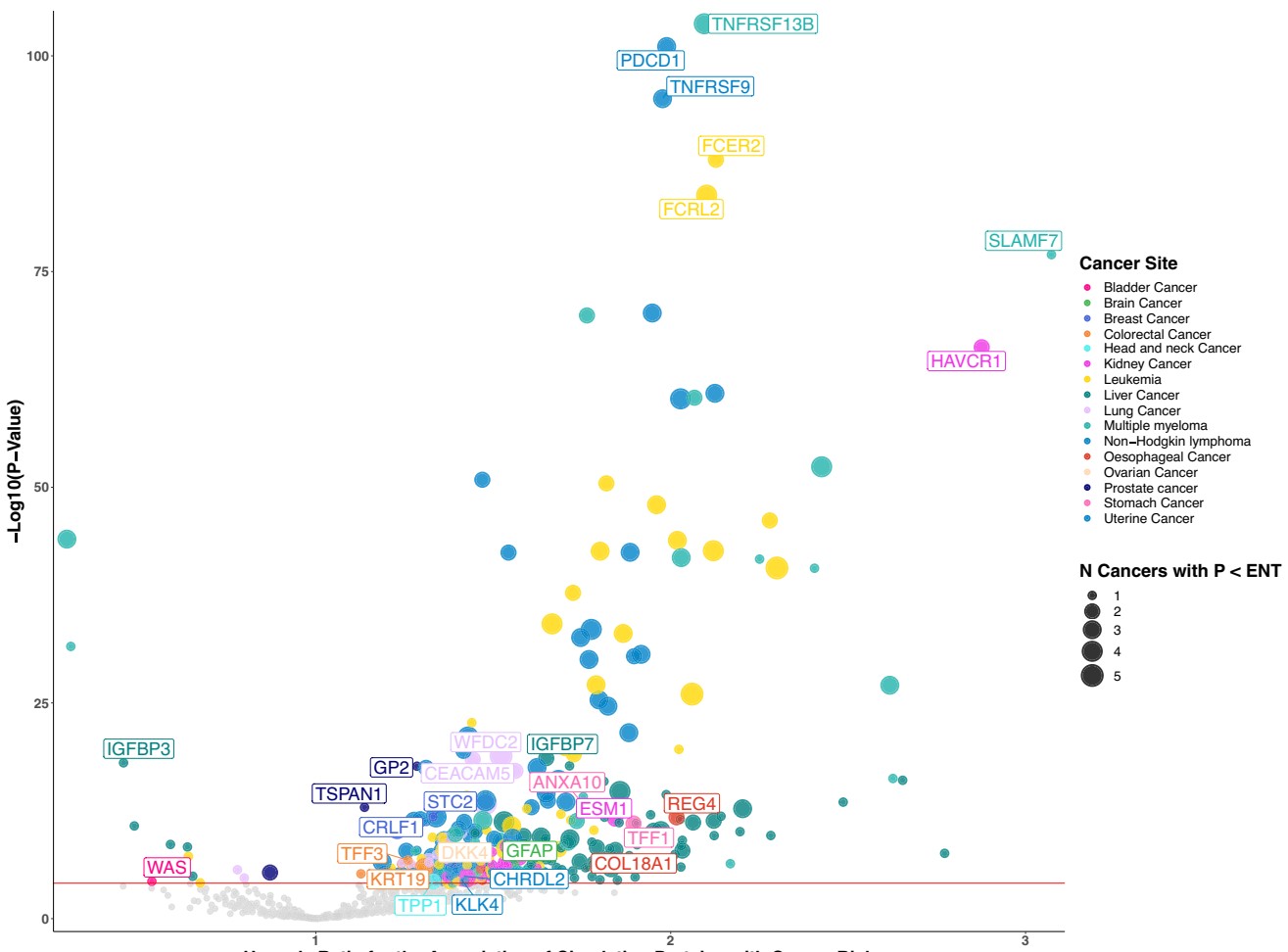

**Fig. 2 | Volcano plot for the prospective associations of circulating proteins with risk of cancer.** Volcano plot displaying the results from the prospective observational analyses of 1463 proteins with cancer risk. Top protein-cancer associations plotted with point size indicating the number of ENT significant protein-cancer associations. The point colour represents the cancer site. Hazard ratios per SD for cancer risk are plotted on the *x*-axis while −log₁₀ *p*-values are plotted on the *y*-axis. Protein names and hazard ratios are labelled to highlight a selection of associations significant after correction for multiple testing (*p* < 0.05/ 639). Hazard ratios and 95% confidence intervals for each cancer site were separately estimated using two-sided Cox proportional hazards regression models. N-number, ENT – effective number of tests. Source data are provided as a Source Data file.

[TPP1, 1.33 (1.16−1.52)]. Little evidence for protein associations was observed in these data for cancers of the pancreas, thyroid, lip and oral cavity, or melanoma after correcting for multiple tests. Limited heterogeneity was observed after stratifying the protein-cancer associations by sex, however, none survived multiple testing corrections (Supplementary Data 4). Pathway analyses for ENT-significant protein-cancer associations, grouped by cancer endpoint, highlighted a potential role for the adaptive immune response in haematological cancers (Supplementary Fig. 3–5). Further adjusting for time since the last meal did not materially affect the magnitude and precision of the ENT significant associations (Supplementary Data 8).

**Analysis stratified by the time between blood draw and diagnosis**
In stratified analyses, we identified 107 of the 618 ENT significant protein-cancer associations as ENT significant in the analysis of cases diagnosed more than seven years after blood draw, representing 72 unique proteins [cancers of the blood: 14, liver: 13, lung: 11, stomach: 5, breast: 3, oesophagus: 3, kidney: 2, colorectum: 1, prostate: 1, thyroid: 1] (Fig. 3). Among the proteins associated with risk of haematological cancers, we identified associations with risk of multiple blood cancers for members of the FC-receptor protein [FCRL1, FCRL2, FCRL3, FCRL5,

FCRLB] and TNF receptor families [TNFRSF4, TNFRSF9, TNFRSF13B, TNFRSF13C, TNFSF13B, TNFSF13]. Among the 618 ENT significant protein-cancer associations, 398 were also ENT significant in the analysis of cases diagnosed within three years of blood draw, representing 256 unique proteins [cancers of the blood: 193, liver: 15, lung: 18, colorectum: 12, kidney: 7, prostate: 6, stomach: 3, bladder: 1, oesophagus: 7, breast: 1, brain: 1, ovary: 1, head and neck:1], which may indicate effects of reverse causation.

**Integrating existing publicly available datasets on drug targets**
We identified 38 proteins associated with the risk of at least one cancer that was also the target of a drug currently approved and available [haematological malignancies (20), liver (17), kidney cancer (7), oesophageal adenocarcinoma (1), and lung cancer (1)]. Most of these proteins were the target of monoclonal antibodies (21) and small molecule inhibitors (13). The proposed action for most of these drugs would be to reduce the cancer risk as indicated in our observational analyses, i.e., the drug would inhibit a protein positively associated with cancer risk. Nine of these proteins are also the target of drugs currently indicated for the treatment of the cancers identified in our risk analyses. These include Dasatinib (EPHA2), Moxetumomab pasudotox (CD22) and Inotuzumab ozogamicin (CD22) indicated in the treatment

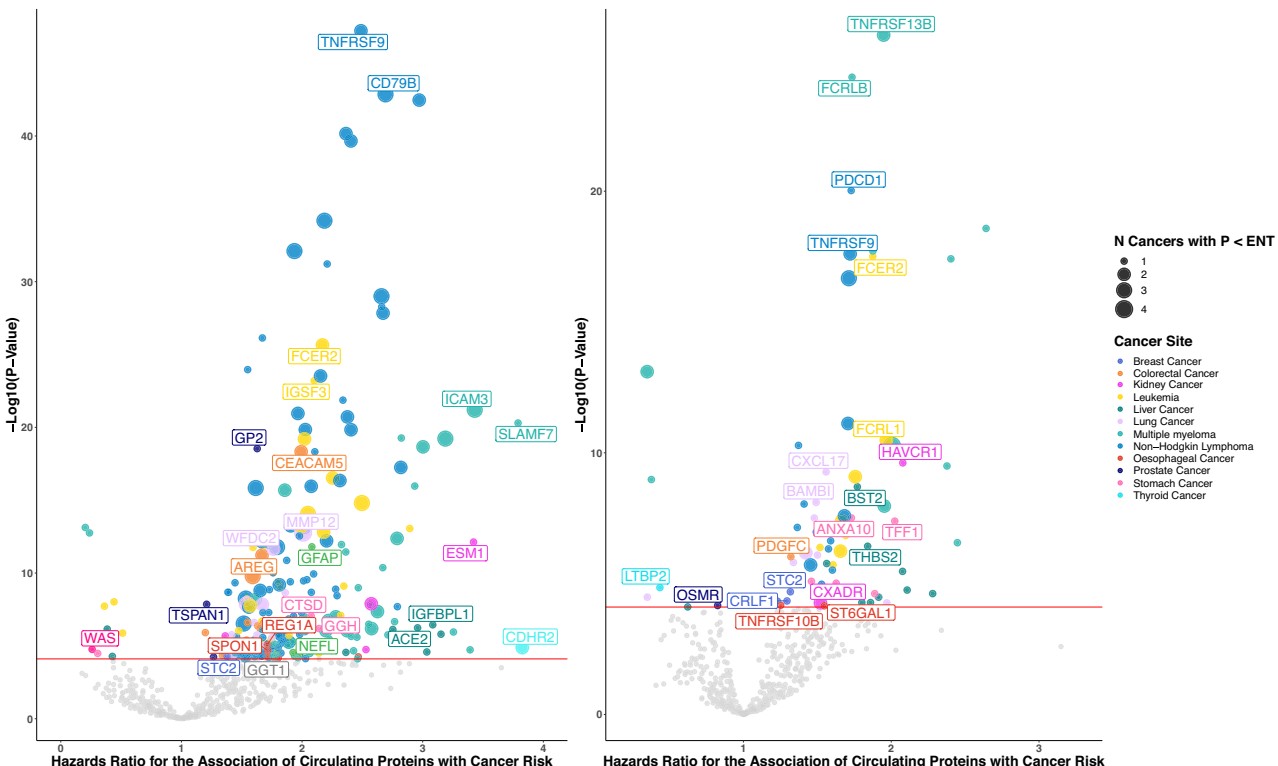

**Fig. 3 | Volcano plots for the prospective association of circulating proteins with risk of cancer by time to diagnosis.** Two volcano plots displaying the results from prospective observational analyses of 1463 proteins with cancer risk stratified by time from blood draw to diagnosis, with analyses among cases diagnosed within three years of blood draw (left) and after seven years of blood draw (right). Top protein-cancer associations plotted with point size indicating the number of ENT significant protein-cancer associations. The point colour represents the cancer site.

Hazard ratios for cancer risk per SD are plotted on the x-axis while $-\log_{10}$ p-values are plotted on the y-axis. Protein names and hazard ratios are labelled to highlight a selection of associations significant after correction for multiple testing ($p < 0.05/$ 639). Hazard ratios and 95% confidence intervals for each cancer site were separately estimated using two-sided Cox proportional hazards regression models. N-number, ENT – effective number of tests. Source data are provided as a Source Data file.

of leukaemia subtypes, Brentuximab vedotin (TNFRSF8), Polatuzumab vedotin (CD79B) and Pembrolizumab (PDCD1) indicated in the treatment of NHL subtypes including DLBCL, Elotuzumab (SLAMF7) indicated in the treatment of multiple myeloma, and Regorafenib (EPHA2, PDGFRA, FGFR2) indicated in the treatment of liver cancers (Supplementary Data 5).

### Circulating proteins with both prospective and single *cis*-variant associations

Using 939 *cis*-pQTL, which represented 294 unique proteins, we investigated 498 of the 618 protein-cancer associations that were identified after multiple tests in the main analyses. Three *cis*-pQTL coding for higher TNFRSF14 were associated with a lower risk of NHL after correction for multiple testing ($p < 0.05/939$ tests based on *cis*-pQTL variants), 1:2559766:C:T [0.85 (0.79–0.91)]; 1:2559503:C:A, [0.85 (0.79–0.91)] and 1:2556714:A:G [0.86 (0.80–0.92)] (Fig. 4). We found evidence to support the potential role of an additional 81 proteins in cancer risk as indicated by 106 protein-cancer associations at $p < 0.05$ which did not meet correction for multiple testing (Supplementary Data 6).

### Circulating proteins with both prospective and exome-score associations

We derived exGS that combined known *cis* and *trans*-pQTLs to predict circulating protein concentrations and assessed their associations with cancer risk. We were able to investigate 533 of the 618 protein-cancer associations across 324 unique proteins. After correcting for multiple testing (0.05/533 exGS tests), we identified 28 associations, including 24 for NHL, 2 for leukaemia (SRP14, TREML2), 1 for

both liver (KRT18) and lung cancer (TNR) (Fig. 4). The strongest association was for SRP14 with leukaemia [1.22 (1.16–1.28)] followed by KRT18 for liver cancer [1.29 (1.18–1.42)], CD1C for NHL [1.11 (1.06–1.16)] and TNR for lung cancer [0.92 (0.89–0.95)]. In addition, we found 115 conventionally significant protein-cancer associations, representing 96 unique proteins (Supplementary Data 7) of which 74 were directionally concordant with the results from the prospective analyses.

### Integrated evidence of protein-cancer associations

We identified four proteins that were both associated with the risk of cancer in the main analyses and had directionally concordant, conventionally significant support from all three additional analyses, i.e., long (>7 years) time-to-diagnosis, *cis*-pQTL, and exGS analyses: SFTPA2 for lung cancer [1.24 (1.14–1.35)], TNFRSF1B [1.28 (1.19–1.37)] and CD74 [1.68 (1.49–1.90)] for NHL and ADAM8 for leukaemia [1.87 (1.69–2.06)] (Fig. 5). In addition, we found genetic and observational evidence supporting the role of 45 unique proteins in the risk of cancer that were associated with cancers of the blood (22 proteins), liver (11), lung (6), kidney (5), colorectum (3), prostate (1) (Supplementary Data 9).

Volcano plots for protein associations with risk of individual cancer types can be found in Supplementary Figs. 7–31.

## Discussion

In this large prospective study of 1463 proteins with the risk of up to 19 cancers, we identified 371 plasma protein markers of cancer risk, including 107 that were associated with cancer diagnosed more than seven years after blood draw and many that also had support from

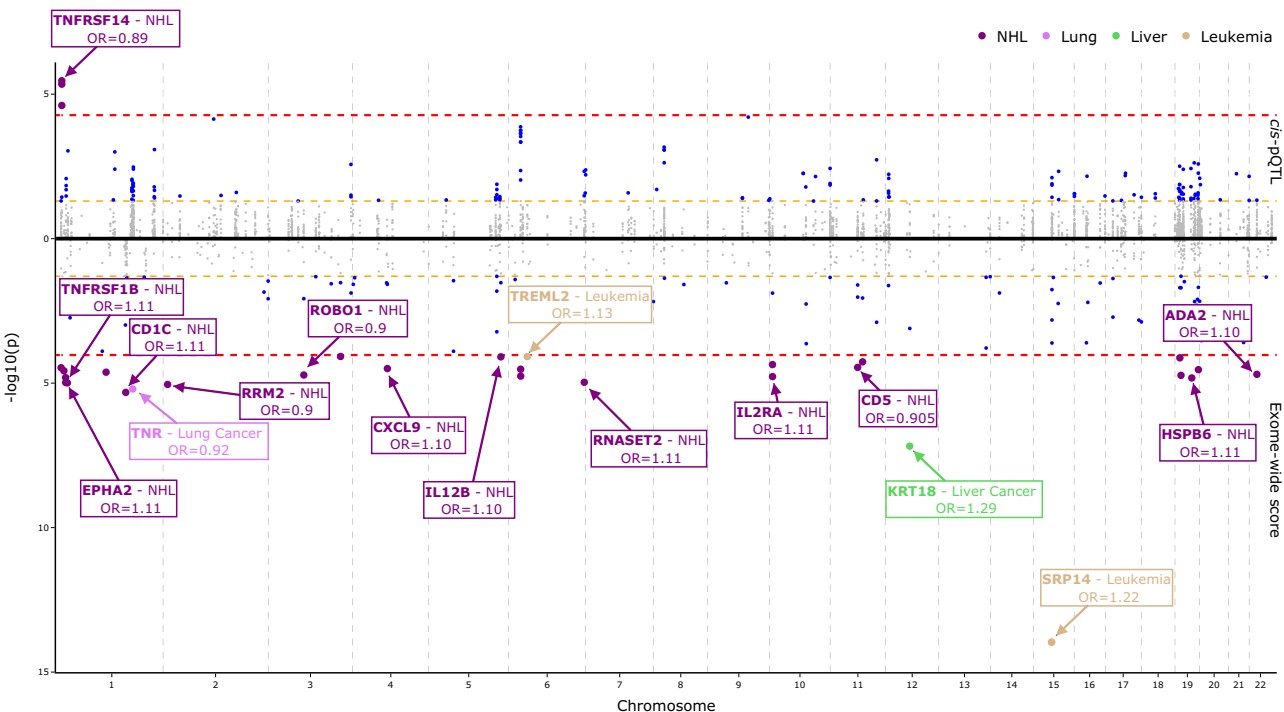

**Fig. 4 | Mirror Manhattan plot for the association of genetically predicted protein concentrations and cancer risk using *cis*-pQTL and exome scores.** This mirror Manhattan plot displays the results of each *cis*-pQTL (top) in the full exome-sequencing cohort within the UK Biobank across European samples for proteins passing correction for multiple testing in the observational results on cancer risk. The y-axis represents the -log₁₀ *p*-values. The bottom of this plot contains the exome-wide score results for genetically predicted proteins. Markers coloured in grey represent results that did not reach the conventional *p* < 0.05 significance threshold, while markers in blue represent conventionally significant results. If a *cis*-variant or an exome-wide score passes Bonferroni significance, those markers are coloured by the cancer site of association. Odds ratios were estimated using logistic regression models to investigate the association of each genetically predicted protein with cancer risk per standard deviation increase. *Cis*-variants were adjusted to be on the same scale. cis-pQTL - cis protein quantitative trait loci, NHL – Non-Hodgkin lymphoma. Source data are provided as a Source Data file.

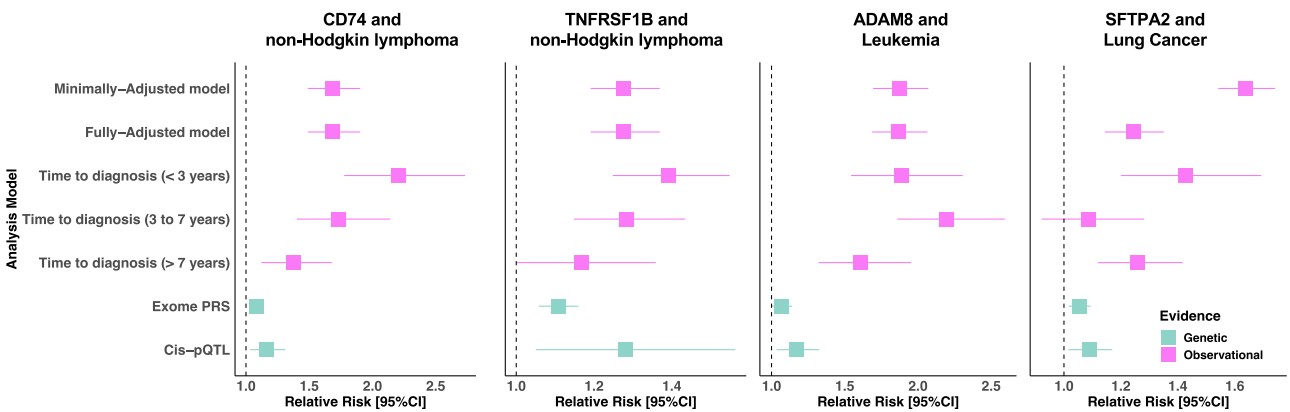

**Fig. 5 | The prospective and genetic associations of SFPTA2 with lung cancer risk, CD74 and TNFRSF1B with risk of non-Hodgkin lymphoma, and ADAM8 with risk of leukaemia.** Plots show the associations for the four proteins that were associated with the risk of cancer in the main analyses and that had directionally concordant, conventionally significant support from all three additional analyses, i.e., long (>7 years) time-to-diagnosis, *cis*-pQTL, and exGS analyses. For each protein–cancer association evidence for the association of concentrations with cancer risk is presented from minimally and fully adjusted models per complementary genetic analyses, which may suggest a role in aetiology. Furthermore, 182 proteins were strongly associated with diagnosis within three years, suggesting potential relevance as biomarkers for early detection. SD, as well as models stratified by time-to-diagnosis, and from exome proteins score and *cis*-pQTL analyses. The observational analyses (minimally adjusted, fully adjusted models, and time-to-diagnosis analyses were conducted in a maximal sample of 44,645 participants, and the genetic analyses were conducted in a maximal sample of 336,823 UK participants. Data are presented as relative risk and 95% confidence intervals. The reference value is 1.0. *cis*-pQTL-*cis* protein quantitative trait loci. Source data are provided as a Source data file.

We identified both proteins that mark common processes across cancer sites and those with associations specific to a particular cancer. The proteins associated with the risk of multiple cancers included GDF15, a stress-regulated hormone that we found to be associated with

an increased risk of eight cancers (liver, aerodigestive and gastrointestinal tract, and haematological malignancies), and MMP12, an enzyme expressed on macrophages that was associated with an increased risk of cancers of the colon, lung, and NHL[15]. However, the majority of protein-cancer associations were cancer site-specific (225 of the 371 proteins), and many also had the majority mRNA expression in the cell or tissue of cancer origin. We note, however, that further evidence for proteins and risk of less common cancers and cancer subtypes may emerge with further follow-up in the UK Biobank or other cohorts.

We found that protein-cancer associations were most prevalent for cancers related to the blood or in tissues with a role in the maintenance of blood composition or with a high throughput of blood, such as the liver, kidneys, and lungs. Further, the smaller number of protein associations for cancers with a higher incidence in this study but whose organs are not directly involved in blood composition (such as breast and prostate) may indicate a more localised effect and highlight the limitation of only measuring blood protein levels when investigating diseases in other tissues. Current multiplex technologies measure a sizable but limited subset of the total human proteome. It is therefore possible that the currently measurable proteins may not include all those with important aetiological relevance for cancer risk. When, in the future, stage and histological grading information becomes available for cancers within the UK Biobank or other cohorts, it may be possible to identify proteins associated with the disease that have progressed beyond the primary organ which may lead to more easily measurable effects in the circulation.

### Integrating prospective observational and genetic evidence for candidate aetiological proteins

We found four proteins associated with cancer that in observational long time-to-diagnosis analyses, and *cis*-pQTL and exGS analyses; CD74 and TNFRSF1B were associated with NHL, and ADAM8 and SFTPA2, were associated with risk of leukaemia and lung cancer, respectively. While each of these three complementary analyses have their own specific biases, the combination of concordant support from all methods may lead to greater confidence for a role in cancer development[10]. Each of these four also appears to have notable biological plausibility. CD74, TNFRSF1B, and ADAM8 all have important roles in the immune system and have enrichment for mRNA expression on candidate cells of origin for NHL and leukaemia. Similarly, SFTPA2 has a well-described role in maintaining healthy lung function and is also majority expressed in alveolar cells, which are a candidate cell of origin for multiple common subtypes of lung cancer[16].

SRP14 was associated with the risk of leukaemia in both observational and exGS analyses and was more strongly associated with the risk of leukaemia in people diagnosed within the first three years. SRP14 has a well-described role in protein targeting in the endoplasmic reticulum, has a high probability of being loss-of-function intolerant (pLi), and is essential for leukaemia and lymphoid malignancy cell survival, as shown using CRISPR knockout models[17]. Notably, the SRP14 exGS association was explained by a single *trans* missense variant (9:5073770:G:T) in JAK2, that leads to constitutively active JAK2, which is known to predispose to various forms of leukaemia[18–20]. Given that *cis*-pQTL did not support a role for SRP14 with leukaemia risk, it is therefore possible that SRP14, as a biomarker of imminent leukaemia diagnosis, may indicate constitutively active JAK2.

Similarly, higher FLT3LG was associated with a lower risk of prostate cancer in both observational and exGS analyses. We found that the FLT3LG exGS was largely explained by *trans*-pQTL that lie in established cancer risk genes involved in the regulation of cell division and DNA repair (CHEK2 [22:28695868:AG:A], ATM [11:108267276:T:C], and TERT [5:1293971:C:T]). For example, carriers of the CHEK2 allele previously reported to increased risk of prostate cancer had lower circulating concentrations of FLT3LG[21,22]. FLT3LG is predominantly

expressed by lymphocytes, in particular natural killer cells, and has a high pLi. It also binds to FLT3, which is expressed on dendritic cells to enhance tumour antigen presentation to facilitate anti-tumour immune responses[23]. Prostate cancer cases carrying high-risk genetic variants in DNA repair pathway genes, such as CHEK2, have a greater risk of progression and are often early onset cases with a higher mutational burden[24,25]. Heightened mutation rates in the absence of effective tumour antigen presentation/immune surveillance would form a coherent biological explanation for higher cancer risk and shorter progression times. Therefore, lower FLT3LG may serve as a potential biomarker of early cancer processes leading to diagnosis among carriers of established prostate cancer risk variants.

Together these findings highlight the need for research into the potential role of blood proteins as circulating readouts that could indicate emerging early carcinogenic processes before diagnosis, and that may complement existing strategies that use germline genetics to identify and monitor *at-risk* populations.

We also identified protein-cancer associations with support from genetic analyses but with a discordant direction of effect. Using *cis*-pQTL, we identified an inverse association of TNFRSF14, a gene with high pLi, with NHL risk, while observational results suggested an association with higher risk, particularly within the initial three years of follow-up. TNFRSF14 is known to acquire loss-of-function mutations early in the development of NHL, which may suggest that it has a protective role during NHL development[26]. TNFRSF14 may therefore be overexpressed as an anti-tumour response to the presence of disease, which could explain our findings. However, current protein assay technology limitations do not enable us to distinguish between multiple proteoforms that may contain higher levels of TNFRSF14 with loss of function variants in these samples.

### Previous studies of proteins and cancer risk

While there have been multiple previous case-control and cross-sectional studies of circulating proteins and cancer risk (with blood taken at or after cancer diagnosis), there are limited published prospective data. We replicate some previously reported prospective associations for proteins and the risk of cancer, which may serve as reassuring confirmation for the reproducibility of findings in this fast-emerging field of multiplex proteomics. We also identified many previously unreported associations possibly due to the prospective study design and/or the large sample size. For example, we replicated the association of CDCP1 with lung cancer risk reported within the EPIC cohort and also found concordant evidence for risk proteins, such as CEACAM5, identified within up to three years before diagnosis in the INTEGRAL project[21,22,27]. We additionally identified risk associations with lung cancer for multiple proteins that were either not previously investigated or that did not meet the significance criteria for multiple testing within previous studies. For colorectal cancer, we were not able to replicate the previously reported associations for several proteins identified in prospective studies using samples taken up to three years before diagnosis or in those studies with relatively modest numbers of incident cases ($n \leq 100$)[28,29]. We also did not replicate protein risk associations previously reported for pancreatic cancer[30]. Nonetheless, our findings are in-line with some of those reported in a cross-cancer case-control study (with blood collected at or after diagnosis) within the Uppsala-Umeå Comprehensive Cancer Consortium biobank; we replicated the reported association of GFAP with glioma and the associations of CNTN5, SLAMF7, MZB1, QPCT and TNFRSF13B with multiple myeloma[31].

Our study has several notable strengths. We examined the role of over one thousand blood proteins in cancer development and reported several hundred novel proteins and cancer associations. The detailed information in the UK Biobank on a wide range of cohort characteristics (including cohort-wide exome data) has made it possible to assess the potential for cancer-specific confounders to

influence our findings and to run complementary genetic analyses on the majority of candidate proteins identified in our observational analyses. Further, information on cancer diagnosis was obtained from data linkage, thus minimising selective dropouts. The cross-cancer approach also reduced outcome selection bias and enabled us to find proteins associated with both multiple and specific cancers, and their subtypes.

Furthermore, the UK Biobank is a mature prospective cohort, which allowed us to assess whether protein-cancer associations were being driven by altered protein levels in individuals who were likely to have preclinical disease at blood draw and/or persisted with longer follow-up. Nonetheless, some haematological cancers can be present long before clinical diagnosis, such as chronic lymphocytic leukaemia[32,33]. Further, liver and kidney disease both have risk factors, including cirrhosis and chronic kidney disease, respectively, that we may expect to perturb the blood proteome far in advance of diagnosis. It is therefore possible that associations with risk observed more than seven years before diagnosis may still be due to either reverse causality or be markers of established risk factors and not aetiological. However, proteins associated with cancer risk long before diagnosis and that have support from complementary genetic analyses may warrant follow-up as potential cancer risk factors.

We also note that we only analysed protein concentrations measured at baseline and therefore were not able to address potential regression dilution bias, which may have led to underestimates of relative risks. Also, while this is the largest cohort study of plasma proteins and cancer to date, we had relatively limited power to detect protein-cancer associations for less common cancer sites and subsites that nonetheless hold substantial public health importance. Finally, the UK Biobank predominantly consists of adults of White ethnicity and who have a more favourable risk profile compared to the national UK population[34]. Proteomics holds significant promise for developing future cancer prevention initiatives that are needed to address the predicted increase in cancer burden among diverse populations, and so further studies into the proteomics of cancer risk including in non-White populations are necessary[35]. This is especially important as the limited observational and genetic evidence suggests that inherited determinants of proteins and the protein-cancer associations can vary between populations of different ancestry[27,36–39].

Several research priorities are leading from our findings that are necessary to pursue to more fully understand the roles of proteins in cancer development and progression. These priorities include more large-scale prospective data from mature cohorts, such as in the European Investigation into Cancer and Nutrition (EPIC), to replicate our findings, and further complementary genetic studies, including Mendelian randomisation analyses. As new GWAS data for cancers of the blood, liver, and kidney become available, further investigations into aetiology using genetic epidemiology will be possible. Where protein associations prove replicable, it will be necessary to better understand their role at the tissue and cellular level. This is of particular interest given proteins are the target of 98% of all drugs and that 38 of our candidate aetiological proteins are the target of existing drugs, of which nine had further directionally concordant evidence from genetic analyses supporting their role in cancer development[40]. Nonetheless, substantial additional research would be needed to assess any potential for therapeutic prevention, including functional and experimental studies, and those to assess potential toxicity.

In conclusion, we discovered multiple associations between blood proteins and cancer risk. Many of these were detectable more than seven years before cancer diagnosis and had concordant evidence from genetic analyses, suggesting they may have a role in cancer development. We also identified proteins that may mark early cancer processes among carriers of established cancer risk variants, which may serve as potential biomarkers for risk stratification and early diagnosis.

## Methods

### Observational data

**Ethical approval.** The study was approved by the National Information Governance Board for Health and Social Care and the National Health Service Northwest Multicenter Research Ethics Committee (06/MRE08/65).

**Study population.** This study is based on data from the UK Biobank participants, a prospective cohort of 503,317 adults aged between 39 and 73, recruited between 2006 and 2010 from across the UK. The study design and rationale have been described elsewhere[34,41]. Briefly, eligible participants were those registered with the National Health Service in England, Scotland or Wales who lived within travelling distance of one of the 22 assessment centres in these regions. In total, ~5% of invited participants joined the study by attending a baseline visit, where they completed a touchscreen questionnaire, had anthropometric data and biological samples taken by trained staff, and gave informed written consent to be followed up through national record linkage.

**Exposure and outcome assessment.** Non-fasting blood samples were collected from all participants at recruitment and plasma was prepared and stored at −80 °C. Protein measurements were generated using the Olink Proximity Extension Assay in 54,306 participants selected as part of the UK Biobank Pharma Proteomics Project (UKB-PPP). Samples were selected for inclusion in the UKB-PPP based on a number of factors described in detail elsewhere[38]. In brief, an initial 5500 were pre-selected by UKB-PPP members. A further 44,502 representative participant samples were selected from the UK Biobank, stratified by age, sex, and recruitment centre. The remaining samples were chosen as part of a second-picking process based on a variety of criteria including membership of a COVID-19 case-control imaging study. Plasma samples were transferred to the Olink Analyses Service, Uppsala, Sweden for measurements.

Olink assay technology and analyses are described in detail elsewhere[42]. In brief, the relative abundance of 1463 proteins was quantified using antibodies distributed across four 384-plex panels: inflammation, oncology, cardiometabolic, and neurology. Blood samples were assayed in four 384-well plates consisting of four abundance blocks for each of the four panels per 96 samples using the Olink Explore platform, which is based on proximity extension assays (PEA) that are highly sensitive and reproducible with low cross-reactivity. Relative concentrations of the 1463 unique proteins were read out by next-generation sequencing. Measurements are expressed as normalised protein expression (NPX) values that are log-base-2 transformed. Protein values below the limit of detection (LOD) were replaced with the LOD divided by the square root of 2 and each protein was rescaled to have a mean of 0 and a standard deviation (SD) of 1[27]. Protein values were subsequently inverse rank normal transformed.

Cancer registration and death data were obtained through record linkage to national registries (NHS Digital for England and Wales using participants' NHS numbers, and NHS Central Register for Scotland using the Community Health Index). Data were available until the censoring date (December 31, 2020, in England and Wales and November 30, 2021, in Scotland) or until participants died, withdrew consent for future linkage or were reported to have left the United Kingdom. Further information on data linkage is available from https://biobank.ndph.ox.ac.uk/crystal/crystal/docs/CancerLinkage.pdf). For the observational analyses, the endpoints were defined as the first incident cancer diagnosis, or cancer first recorded in death certificate if there was no previous record of a cancer diagnosis [all coded using the 10th revision of the World Health Organisation's International Statistical Classification of Diseases (ICD-10)]: head and neck (C00–14, C32), oesophagus (C15), stomach (C16), colorectum (C18–20), liver (C22), pancreas (C25), lung (C34), malignant melanoma (C43), breast

in women (C50), uterine (C54), ovary (C56), prostate (C61), kidney (C64–65), bladder (C67), brain (C71), thyroid (C73), and the blood cancer subgroups non-Hodgkin lymphoma (NHL; C82–85), multiple myeloma (C90), and leukaemia (C91–95). The following subclassifications of these cancer groupings were also considered: oral (C00–14) and lip and oral cavity (C00–06) within head and neck cancers (C00–14, C32); adenocarcinoma of the oesophagus (C15, morphology codes ICD-O-3 8140–8573) within oesophageal cancer (C15); colon (C18) and rectum (including rectosigmoid junction, C19–20) within colorectal cancer (C18–20); adenocarcinoma of the lung (C34, morphology codes ICD-O-3 8140, 8211, 8250–8260, 8310, 8323, 8480–8490 and 8550), squamous cell carcinoma (C34, morphology codes ICD-O-3 8070-8072), small cell carcinoma (C34, morphology codes ICD-O-3 8041-8042) within lung cancer (C34); and diffuse lymphoma (C83) within NHL (C82–85). The person-years of follow-up were calculated from the date of recruitment until the date of first registration of malignant cancer, death due to cancer, death, loss or end of follow-up, or censoring date, whichever came first.

### Exome-sequencing in the UK Biobank and exonic pQTL discovery

Exome-sequencing data preparation and quality control procedures in the UK Biobank have been previously described[43]. In brief, exome capture was done using the IDT xGen Exome Research Panel v1.0 that underwent 75 bp paired-end Illumina sequencing on the NovaSeq 6000 platform using the S2 and S4 flow cells. BWA-MEM was used to map reads to GRCh38 with variant calling performed by DeepVariant using a 100 bp buffer at each site of the custom target regions. We extracted 27,335 exome variants associated with circulating protein concentrations on the Olink Explore panel at $p < 5 \times 10^{-8}$ reported by Dhindsa et al. for 50,829 UK Biobank participants[44]. The exome variants reported by Dhindsa et al. underwent a different pipeline using AstraZeneca's Genomics Research (CGR) bioinformatics pipeline[44]. Single Nucleotide variants (SNV) and small insertions and deletions (INDEL) were additionally annotated to SnpEFF v4.3 against Ensembl Build 38.92[45].

### Exclusion and inclusion criteria

Olink proteomics was measured in EDTA plasma on 54,221 UK Biobank participants. Two participants were removed due to withdrawn consent from the UK Biobank and 1429 samples were removed as part of quality control procedures. These procedures included identifying and excluding participants with a median protein value or median interquartile range across all protein values that lay more than five standard deviations outside the scaled mean median protein value or mean median interquartile range across all protein values in the cohort. More details can be found in a previous publication by the UKB-PPP[46]. We further excluded 2969 participants due to cancer diagnosis at or before baseline (except non-melanoma skin cancer C44), 37 whose self-reported sex did not match their genetic sex, 242 who had missing information on height or weight, 2113 who were currently using hormone replacement therapy or oral contraceptives, and 2709 who reported having diabetes at baseline. Following these exclusions, the maximal analysis cohort included 44,645 participants (see Supplementary Fig. 1 for participant flowchart).

### Statistical analysis

**Observational analyses.** All analyses were conducted using Stata release 18.1 and R version 4.1.2. We estimated hazard ratios (HRs) and 95% confidence intervals (CI) for each cancer site separately using Cox proportional hazards regression models with age as the underlying time variable. Missing data in covariates were handled by assigning participants to an "unknown" category for each respective variable. The minimally adjusted models were stratified by age group at recruitment (< 45, 45–49, 50–54, 55–59, 60–64, and ≥ 65 years) and self-reported gender where applicable and adjusted for geographical

region (London, North-West, North-East, Yorkshire and Humber, West Midlands, East Midlands, South-East, South-West, Wales, and Scotland), and Townsend deprivation index (fifths, unknown). Multivariable-adjusted models were additionally adjusted for cancer-specific risk factors (see Extended "Methods"). Cancer-specific risk factors were chosen upon review of the literature and restricted to variables available in the UK Biobank. We used an effective number of tests (ENT) correction for multiple testing, applied in a family-wise manner by cancer type. The ENT method accounts for multiple tests by applying a Bonferroni correction that determines the number of independent tests as the number of principal components needed to explain 95% of the variance in protein abundance. In this case, this was 639 independent tests[27].

We examined protein and cancer-risk associations by time to diagnosis (diagnosed in < 3 years, 3–7 years, and > 7 years of follow-up) to investigate the potential effects of reverse causality. We also conducted a sensitivity analysis by self-reported sex (women and men) to investigate potential sex differences for protein-cancer associations that passed multiple testing corrections. We tested the heterogeneity of risk coefficients between the subgroups in each stratified analysis using inverse variance weighting, testing for statistical significance with a χ2 test with k-1 degrees of freedom, where k is the number of subgroups. Finally, we further adjusted the 618 ENT significant risk-factor adjusted protein-cancer associations for time since the last meal to investigate the potential impact of fasting status.

### Integrating existing publicly available datasets on gene expression

To provide greater biological context for identified protein-cancer associations, we extracted single-cell RNA expression from the Human Protein Atlas to describe mRNA expression in cancer-free individuals for genes that code for the identified protein markers in our main observational analyses[47]. Normalised expression levels were extracted for genes in 30 different human tissues and 82 cell types. Gene expression specificity at the cell or tissue type level was calculated as the ratio of each gene cell type or tissue expression to the total expression of each gene across all cell or tissue types. We subsequently grouped genes into majority expression (more than 50% of total expression in each cell or tissue type) and enriched expression (between 10% and 50% of total expression in each cell type or tissue). For proteins with either mRNA enriched or majority expression in at least one cell or tissue type, we also mapped these to their likely candidate cell and tissue of origin where possible.

### Integrating existing publicly available datasets on drug targets

We gathered information on the potential *druggability* of proteins with evidence of a cancer risk association in our main analyses by extracting information on whether a protein was the target of a known drug from the Open Targets Platform[48]. Subsequently, we filtered information from Open Targets to identify drugs that were approved and on the market by additionally cross-referencing against the ChEMBL database and other drug databases including DailyMed and the Electronic Medicines Compendium[49–51]. Proteins identified as the target of an available drug were additionally annotated with information on whether the effect of the drug would act to reduce or increase the proposed protein association with cancer risk.

### Cis-pQTL and exome-wide genetic score on cancer outcomes

We further investigated protein-cancer associations identified after correction for multiple testing in observational analyses using two genetic approaches: single *cis*-pQTL risk analyses, where *cis*-pQTL were available for the protein of interest, and using an exome-wide genetic score approach. No exonic variants were identified by Dhindsa et al. for PREB, ING1, NPM1, PQBP1, SEPTIN9, KRT14 and ARTN and so were not considered in these analyses. In all exome-wide analyses, variants were

oriented to the protein-increasing allele and exGS were calculated by summing the number of independent (clumping r$^2$ < 0.05, 10,000KB) protein-increasing alleles, weighted by betas reported in Dhindsa et al., and projected in up to 337,543 European UK Biobank participants with exome-sequencing (Supplementary Data 10) using PLINK2[52]. We subsequently used logistic regression models to estimate the association of each genetically predicted protein with cancer risk, using both *cis*-pQTL and exGS models, for each protein-cancer association identified in observational analyses. Models were adjusted for age, sex, and the first 10 genetic principal components of ancestry. For sex-specific cancers (breast, prostate, ovary and uterine), sex was excluded from the model. *Trans*-pQTL single variant analyses were conducted to contextualise which genes may drive protein associations with cancer risk from exGS analyses. In addition, we annotated exGS and single variant analyses with pLi from gnomAD and used IntOGen to annotate driver genes[26,53]. In the exome analysis, conventional significance was defined as *p* < 0.05, while Bonferroni correction was used as the threshold for multiple test correction across the number of *cis*-pQTL or exGS analysed for *cis*-pQTL or exome-wide genetic scores, respectively.

### Combined evidence from prospective observational and genetic analyses

To enhance our understanding of a protein's likelihood of having a role in cancer aetiology, we combined evidence from observational long time-to-diagnosis analyses (> 7 years between blood drawn and diagnosis), *cis*-pQTL analyses, and exGS analyses, and categorised protein-cancer associations by degree of directionally concordant support from each of these three analyses. Acknowledging that not all proteins may have *cis*-pQTL, we ranked proteins as most likely to be aetiological risk factors if all three types of analyses supported an association at conventional significance, followed by long time-to-diagnosis and *cis*-pQTL analyses, then long time-to-diagnosis and exGS, exGS and cis-pQTL, and finally any one of long time-to-diagnosis, *cis*-pQTL, or exGS analyses.

### Reporting summary

Further information on research design is available in the Nature Portfolio Reporting Summary linked to this article.

## Data availability

This research has been conducted using the UK Biobank Resource under Application Number 67506. Researchers can apply to use the UK Biobank resource for health-related research that is in the public interest (https://www.ukbiobank.ac.uk/register-apply/). We wish to express our gratitude to the participants and those involved in building the resource. Source data are provided with this paper.

## Code availability

The code and weights for the protein profiles generated in this study can be found:https://github.com/GenomicEPIOX/paper_1463_proteins_19_cancers_UKBB/tree/main.

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

## Acknowledgements

This work was supported by Cancer Research UK (grant numbers C8221/A29017 and C8221/A29186) to fund the centralised pooling, checking and data analysis. Keren Papier is supported by Wellcome, Our Planet Our Health (Livestock, Environment and People—LEAP) [grant number 205212/Z/16/Z]. Tammy Tong is supported by a UK Research and Innovation Future Leaders Fellowship (MR/X032809/1). Trishna Desai is supported by a Cancer Research UK studentship (grant number C8221/A30904). Chibuzor F. Ogamba is supported by an NDPH studentship. The funders had no role in study design, data collection, analysis, decision to publish, or preparation of the manuscript.

## Author contributions

These authors contributed equally: Keren Papier, and Joshua R Atkins These authors jointly supervised this work: Karl Smith-Byrne, Ruth C Travis Study concept and design: K.P., J.R.A., K.S.B. and R.C.T. Statistical analysis and draughting of the initial manuscript: K.P., J.R.A. and K.S.B. Interpretation of the data, critical revision of the manuscript for important intellectual content, and approval of the final submitted version: K.P., J.R.A., T.Y.N.T., K.G., T.D., C.F.O., M.P., G.K.R., I.G.M., T.J.K., K.S.B. and R.C.T.

## Competing interests

The authors declare no competing interest.
