## [Peer Review File · Nature Communications]

Identifying proteomic risk factors for cancer using prospective and exome analyses: 1,463 circulating proteins and risk of 19 cancers in the UK BiobankREVIEWER COMMENTS

Reviewer #1 (Remarks to the Author): expertise in proteomics risk analysis

Authors have undertaken a multi-omics, cross-cancer approach to comprehensively assess the plasma proteome in relation to cancer risk. This manuscript represents a large undertaking and is important work, but the reviewer requires major revisions before it is suitable for publication. The reviewer's primary concern is that no validation was performed. Since authors have more than adequate sample size, the reviewer requests the cohort be split into a primary and validation dataset to see if the main findings still hold. Validation in other datasets should also be considered. Other comments are below.

- Line 106: These are non-fasting blood samples. Authors should show data or cite something to prove fasting status will not impact the presented results.
- Line 122: Where are these protein values? Some summary statistics are shown in supplemental. The full dataset should be made available in supplemental and deposited to appropriate publicly accessible repository.
- Line 123: The reviewer is not familiar with the UK national registries. Is this data publicly accessible inside the UK? Outside? Please provide some more details so readers can acquire this data or know the process for acquiring this data if they wanted.
- Line 129: First cancer diagnosis is mentioned. Were individuals with multiple cancers considered in the analyses? If so, please show that the results are similar if patients with multiple cancers are not considered.
- Line 143: Although authors mention it was previously described, they should specify how many samples were used for sequencing. Was this for all 54,306 patients? Are these the same blood/plasma samples mentioned in the previous section? PBMCs? What kind of quality controls were done on these samples by the cited authors? What quality controls were done by the authors? Is batch information available for these samples? Authors should verify no batch effects.
- Line 144: Authors should make sequencing data available through a protected access repository to enable reproduction of results.
- Line 145: Citing a website is not acceptable. Further, the link does not even work. This needs to be rectified by authors. Please check to make sure all references of "...as previously described..." are to peer reviewed publications.
- Line 155: What kind of quality control? Authors should be more explicit to enable replication of findings.
- Extended methods: some models were not adjusted for smoking including stomach and breast cancer, but the reviewer is familiar with literature showing increased risk of stomach and/or breast cancer from smoking. The modeling should be redone taking smoking status or pack years into account. If this information is not available for all cancer sites, then authors should acknowledge this as a weakness in the discussion.
- Extended methods: Extended figure 1 is very helpful. Consider including this as a panel in the main manuscript.
- Line 247: Figure 1: Given the breadth and scope of the data, the reviewer was a little disappointed that the first figure was entirely hazard ratios and p-values. The first figure is a great opportunity to "shock and awe" readers and represent the data in a global/multi-variate manner. Authors should take inspiration from CPTAC and TCGA papers, which have some excellent data visualizations of their cohorts.
- Line 247: Figure 1: These volcano plots are fairly busy and cluttered. Are there common

pathways/gene sets amongst the significant genes? The reviewer would like to see pathway enrichment of significant genes (and possibly have authors move these volcanos to supplemental). This would give the opportunity to show pathway analysis plots of the top 5-10 pathways instead of showing the top dozens of genes for each cancer. A heatmap of pathway enrichment scores/p-values with the y axis being pathways and the x axis being cancer type could help readers see commonalities/differences amongst the disease sites.

- Line 247: Figure 1: There are clearly separate distributions among some of the disease sites. Breast cancer looks distinct from endometrial and ovarian cancers for example. The reviewer would like these to be plotted/shown separately.

- Line 250: Figure 2 seems very out of place. The axes are hard to read. The data in the heatmap was not clustered. What do white values mean? The caption does not give enough information. What is this figure trying to show? Is this the publicly available data mentioned in the methods? The figure does not reflect the caption 'evidence for cellular and tissue enrichment of mRNA expression for cancer risk proteins'. Were any comparisons performed that can be supported with a p-value to claim enrichment? What do authors mean by 10% and 50% of total expression? A 1.1 increase and 1.5 increase, respectively? These seem arbitrary thresholds to use as evidence of enrichment.

- Line 284: Figure 3: These volcano plots are incredibly busy and hard to read. A bar plot of hazard ratios colored by cancer site with *'s for the level of significance would be much easier to interpret.

- Line 291: Are the authors proposing these circulating markers as putative drug targets? This seems like a big stretch since it is unclear if the changes in the circulating markers are based on systemic differences in the patients or if they are tumor-specific.

- Line 305: Why 498 out of 621?

- Line 309: The reviewer likes this plot. Why do authors think there so many hits for NHL compared to the others? Blue and grey points need to be labeled in the legend.

- Line 314: Why 533 out of 621?

- Misc comment: The figures are missing some panel labels. Mentions of figures in the text need to specify the panel.

Reviewer #2 (Remarks to the Author): expert in epidemiology and cancer risk

Noninvasively collected biospecimen are useful sources for cancer risk assessment, disease stratification and generating risk-prediction models. This very well written and appropriately designed epidemiology association study provides timely and health relevant information about protein-based cancer detection. Methodology and interpretation of results is excellent.

A few comments are:

Since most of the samples are from Caucasian population, how much this information would be applicable to other ethnic and racial groups?

It would be useful to include information about any plan to validate some identified protein markers. Any reason for identifying a small number of biomarkers for few cancer types?

Reviewer #3 (Remarks to the Author): expertise in epidemiology

This paper is a study that confirmed the association between plasma proteins and cancer risk in 44,645 people from the UK biobank. The validity of the protein marker is partially proven using a

method to confirm the protein marker through cancer risk analysis of QTL/exGS among all biobank participants. There are currently few such studies, so it is not possible to confirm their external validity. In this study, pathogenic protein markers were selected using exome score and expression information, and we believe this method is the most appropriate method at present. Future research should be conducted to confirm whether the protein markers identified in this study are the same in other population studies, that is, to confirm external validity.

In this study, eligible criteria of subjects were appropriate and the analysis method was well described. The analysis of markers was also well described, and it was also appropriate to set a period of > 7 years (from blood collection to cancer diagnosis) to see the effect of protein markers on cancer development. Thus the proteins ascertained in this study are thought to be proper markers in prediction of future cancer development.

The exome genetic score and protein profile you analyzed were not presented in the supplementary. Please include the UK biobank website where such profiles are presented in the methods of this paper. Please include the UK biobank website where such profiles are presented in the methods of this paper for reference by other researchers.

Response to reviewers

Re: Identifying proteomic risk factors for cancer using prospective and exome analyses: 1,463 circulating proteins and risk of 19 cancers in the UK Biobank

We thank the reviewers for their comments, these have greatly improved our manuscript. We have addressed the reviewers' comments point-by-point below. We present reviewers' comments below, followed by our responses (indented). Revised manuscript text appears as highlighted text.

Reviewer reports	2
Reviewer #1	2
Reviewer #2	11
Reviewer #3	13
References.....	15

Reviewer reports

Reviewer #1

Authors have undertaken a multi-omics, cross-cancer approach to comprehensively assess the plasma proteome in relation to cancer risk. This manuscript represents a large undertaking and is important work, but the reviewer requires major revisions before it is suitable for publication.

1. The reviewer's primary concern is that no validation was performed. Since authors have more than adequate sample size, the reviewer requests the cohort be split into a primary and validation dataset to see if the main findings still hold. Validation in other datasets should also be considered. Other comments are below.

Authors' Response:

We thank the reviewer for their comment and agree that validation is an important consideration when conducting research into proteomics and cancer incidence. There are multiple methods that can be used to approach the validation of an identified protein-cancer association that can include: triangulation using genetic methods that leverages favourable absence of many common biases and sources of confounding present in prospective cohort study designs; experimental and functional studies aimed at elucidating the potential mechanisms and sites of action for protein-cancer associations; and also the replication of prospective results using external and similarly designed cohort studies. As part of our current study, we have included a large complementary exome-based analysis for the proteins with genetic determinants and that were associated with cancer risk in the prospective analyses.

As acknowledged by Reviewer #3, replication in other large prospective studies is not currently possible due to the lack of similar large-scale proteomic resources currently available on a similar breadth of cancer outcomes. In terms of whether or not to use a split-sample method for validation within the UK Biobank cohort, the sample size in the current analysis is not large enough across each cancer outcome to allow for a robust discovery-replication design given the modest number of cancer cases for most outcomes in this study.

We include below a summary of results from existing studies of blood proteomics and cancer risk; one of these studies was a case-case study including multiple cancers [1], while the other was a nested case-control study designed specifically to identify markers for the early detection of lung cancer [2]. Therefore, neither study was designed to identify aetiologically relevant markers for cancer, and so neither are suitable for direct validation of our findings. Nonetheless reasonable concordance is found for our main findings in terms of significance and direction of effect (Please see figures below).

We look forward to results from future studies of the circulating proteome and cancer risk in prospective studies that may additionally generate observational replication for these associations.

2. Line 106: These are non-fasting blood samples. Authors should show data or cite something to prove fasting status will not impact the presented results.

Authors' Response:

We thank the reviewer for their comment on fasting status and its potential impact on the circulating proteome. In our sensitivity analyses that included additional adjustment for time since last meal, the protein-cancer associations were materially unchanged. We have now included these results in Extended Table 8, and describe the results in our revised text.

Methods, p.5, Lines 181-182

Finally, we further adjusted the 618 ENT significant risk-factor adjusted protein-cancer associations for time since last meal to investigate the potential impact of fasting status.

Results, p.7, Lines 278-281

Further adjusting for time since last meal did not materially affect the magnitude of the ENT significant associations (Extended Table 8).

3. Line 122: Where are these protein values? Some summary statistics are shown in supplemental. The full dataset should be made available in supplemental and deposited to appropriate publicly accessible repository

Authors' Response:

Relative quantification of 1,463 protein concentrations was generated using Olink technologies in plasma samples from participants whose data were analysed in this manuscript as part of the Pharma Proteomics Project within the UK Biobank. These data are available following an approved application to the UK Biobank (*found at <https://www.ukbiobank.ac.uk/register-apply/>*, see response to comment 4 below) and are not data that were generated or that are owned by the authors.

4. Line 123: The reviewer is not familiar with the UK national registries. Is this data publicly accessible inside the UK? Outside? Please provide some more details so readers can acquire this data or know the process for acquiring this data if they wanted.

Authors' Response to 3 and 4:

The UK Biobank is a large cohort study containing de-identified lifestyle, genetic, and health information (including record linkage to national cancer registries) on 500,000 participating adults from across the UK. All bona fide researchers can apply to use the UK Biobank resource for health-related research that is in the public interest (<https://www.ukbiobank.ac.uk/register-apply/>). We have now included this information in our revised text.

Data access, p.11, Lines 460-463

This research has been conducted using the UK Biobank Resource under Application Number 67506. All bona fide researchers can apply to use the UK Biobank resource for health-related research that is in the public interest (<https://www.ukbiobank.ac.uk/register-apply/>). We wish to express our gratitude to the participants and those involved in building the resource.

5. Line 129: First cancer diagnosis is mentioned. Were individuals with multiple cancers considered in the analyses? If so, please show that the results are similar if patients with multiple cancers are not considered.

Authors' Response

We seek to identify proteins that may be aetiological risk factors for individual cancers and so investigated associations for first incident cancer diagnosis for each cancer type. We agree that looking at multi-morbidity would also be informative, but that this is beyond the scope of the current study.

6. Line 143: Although authors mention it was previously described, they should specify how many samples were used for sequencing. Was this for all 54,306 patients? Are these the same blood/plasma samples mentioned in the previous section? PBMCs? What kind of quality controls were done on these samples by the cited authors? What quality controls were done by the authors? Is batch information available for these samples? Authors should verify no batch effects.

Authors' Response

In UK Biobank, more than 400,000 individuals were sequenced and it was a subset of these individuals who had Olink proteomics measured in EDTA plasma from the same blood sampling that was used to derive the PBMCs from which the DNA was extracted for sequencing. Quality control included the removal of samples with assay warnings or likely sample swaps as well as the identification and removal of extreme outlier samples as described in response to comment 9 below.

Minimal batch or plate effects (over 99% of proteins had variability less than 10% due to plate and less than 1% due to batch) were observed as part of the extensive evaluation performed within the Pharma Proteomics Project for these samples [3] and coefficients of variation were modest across proteins measured (Median: 5.59, IQR (4.73-6.85)).[3]

7. Line 144: Authors should make sequencing data available through a protected access repository to enable reproduction of results.

Authors' Response

All exome sequence data can be accessed through the UK Biobank resource described above (see our response to Comment 4, and information at <https://www.ukbiobank.ac.uk/register-apply/>).

8. Line 145: Citing a website is not acceptable. Further, the link does not even work. This needs to be rectified by authors. Please check to make sure all references of "...as previously described..." are to peer reviewed publications.

Authors' Response

We have checked and confirmed that the citations included in the 'as previously described..' paragraph are all peer reviewed publications. The included link in the Methods section however does not refer to a scientific article but rather to a UK Biobank resource describing the available up to date cancer registry data. This manual is regularly updated by the UK Biobank team and may be useful for other researchers using these data. We are sorry to hear that the link did not work and have verified that it now works.

9. Line 155: What kind of quality control? Authors should be more explicit to enable replication of findings.

Authors' Response

We thank the Reviewer for highlighting the quality control for measurements of Olink proteomics. For clarity we have updated the text to more thoroughly describe the sample selection and quality control processes:

Methods, p.4, Lines 147-152

Olink proteomics were measured in EDTA plasma on 54,221 UK Biobank participants. Two participants were removed due withdrawn consent from the UK Biobank and 1,429 samples were removed as part of quality control procedures. These procedures included identifying and excluding participants with a median protein value or median interquartile range across all protein values that lay more than five standard deviations outside the scaled mean median protein value or mean median interquartile range across all protein values in the cohort. More details can be found in a previous publication by the UKB-PPP [3].

10. Extended methods: some models were not adjusted for smoking including stomach and breast cancer, but the reviewer is familiar with literature showing increased risk of stomach and/or breast cancer from smoking. The modelling should be redone taking smoking status or pack years into account. If this information is not available for all cancer sites, then authors should acknowledge this as a weakness in the discussion.

Authors' Response

We thank the reviewer for their suggestion. We have now updated our stomach and breast cancer analyses to include further adjustment for smoking status, which is now reflected in the manuscript text, tables, and figures. Additional adjustment for smoking status had no impact on breast cancer results while three (GDF15, MMP12, CXCL17) of the previously reported eight proteins associated with risk of stomach cancer did not meet ENT significance in updated analyses.

11. Extended methods: Extended figure 1 is very helpful. Consider including this as a panel in the main manuscript.

Authors' Response

We thank the reviewer for this suggestion. We have integrated pertinent information about the study design in the updated Figure 1 that we now include in response to the comment immediately below. More detailed information on our study design can be found in Extended Figure 1.

12. Line 247: Figure 1: Given the breadth and scope of the data, the reviewer was a little disappointed that the first figure was entirely hazard ratios and p-values. The first figure is a great opportunity to "shock and awe" readers and represent the data in a global/multi-variate manner. Authors should take inspiration from CPTAC and TCGA papers, which have some excellent data visualizations of their cohorts.

Authors' Response

We thank the reviewer for the opportunity to redesign our Figure 1. Our revised figure now incorporates both the study design and a high-level view of our findings, which also includes results from the long-lag time observational and the genetic analyses, as well as select results from pathways analyses included in response to reviewer comments.

13. Line 247: Figure 1: These volcano plots are fairly busy and cluttered. Are there common pathways/gene sets amongst the significant genes? The reviewer would like to see pathway enrichment of significant genes (and possibly have authors move these volcanos to supplemental). This would give the opportunity to show pathway analysis plots of the top 5-10 pathways instead of showing the top dozens of genes for each cancer. A heatmap of pathway enrichment scores/p-values with the y axis being pathways and the x axis being cancer type could help readers see commonalities/differences amongst the disease sites.

Authors' Response

We agree that pathway analyses can be an informative method to contextualise patterns in protein associations with cancer risk. We have run a pathway enrichment analysis for proteins identified as being associated with risk for each cancer using the clusterProfiler package (version 3.18.1). Specifically, we used the enricher() function

to identify set of proteins that were overrepresented in annotations from Gene Ontology (GO) biological processes (BP), cellular component (CC), and molecular functions (MF). Significance was defined using the Benjamini-Hochberg method ($p < 0.05$). The main finding highlighted a potential role for the adaptive immune response in hematological cancers. We now describe these methods in the Supplementary Methods page 4 and results on page 7 together with Extended Figures 3-5. A summary of the pathways for the top cancer sites is also presented in the revised Figure 1.

We have additionally produced one combined volcano plot that summarises the main findings for proteins associated with cancer risk (presented as Figure 2) and now include separate volcano plots that show the protein-risk associations for each cancer outcome in the supplementary materials (Extended Methods and Figures 7-31).

14. Line 247: Figure 1: There are clearly separate distributions among some of the disease sites. Breast cancer looks distinct from endometrial and ovarian cancers for example. The reviewer would like these to be plotted/shown separately.

Authors' Response

We now provide separate volcano plots for each of the cancer-sites in Supplementary Figures 7-31. We have now also revised Figure 1.

15. Line 250: Figure 2 seems very out of place. The axes are hard to read. The data in the heatmap was not clustered. What do white values mean? The caption does not give enough information. What is this figure trying to show? Is this the publicly available data mentioned in the methods? The figure does not reflect the caption 'evidence for cellular and tissue enrichment of mRNA expression for cancer risk proteins'. Were any comparisons performed that can be supported with a p-value to claim enrichment? What do authors mean by 10% and 50% of total expression? A 1.1 increase and 1.5 increase, respectively? These seem arbitrary thresholds to use as evidence of enrichment.

Authors' Response

We thank the reviewer for their comments on Figure 2. Many of the proteins that are measurable in the circulating concentration are highly expressed in specific tissues and unlikely to associate with cancer risk due to their expression in the plasma alone. However, it can be challenging to contextualise this for high throughput analyses. We therefore include the plots in Figure 2 as a high-level descriptive summary of candidate tissues and cells that produce the measured blood concentrations for proteins that association with cancer risk. These figures are based on publicly available data (<https://www.proteinatlas.org/about/download>) that are referenced in Methods on line 186. We have now moved this figure to Extended Figure 6.

16. Line 284: Figure 3: These volcano plots are incredibly busy and hard to read. A bar plot of

hazard ratios coloured by cancer site with *'s for the level of significance would be much easier to interpret.

Authors' Response

We appreciate this feedback on Figure 3 and have increased the label and font sizes to improve its readability. Additionally, we have included a bar chart in Figure 1.

17. Line 291: Are the authors proposing these circulating markers as putative drug targets? This seems like a big stretch since it is unclear if the changes in the circulating markers are based on systemic differences in the patients or if they are tumor-specific.

Authors' Response

We completely agree with the reviewer that the development and approval of drugs for the prevention of cancer is a long and complex process, and not something that falls within the scope of any individual study. However, the inhibition or agonism of some proteins are already the mechanism of action of approved drugs in current use, which can inform the relative ease for the development of and/or likely human efficacy for any potential future therapeutic intervention via either novel or repurposed drug. Additionally, where we identify a protein associated with cancer incidence and where a drug that acts on that protein has been shown in randomised control trials to treat that cancer, this serves as important evidence for the relevance of that protein in carcinogenesis important for that cancer. We do not suggest therefore that proteins associated with cancer are drug targets but rather that these proteins have an extra layer of evidence that may be informative for the future prioritisation of research into cancer prevention.

18. Line 305: Why 498 out of 621?

Authors' Response

Line 305 refers to the description of results from our study that used genetic variants that lie within or nearby to the gene that codes for a protein being predicted. For several reasons, including how highly conserved a protein is, there may not be any cis-acting variants that predict the circulating concentration of a protein. In our study, we had the ability to proxy the associations of 498 of 618 protein-cancer associations using cis-acting genetic variants.

19. Line 309: The reviewer likes this plot. Why do authors think there so many hits for NHL compared to the others? Blue and grey points need to be labelled in the legend.

Authors' Response

NHL is a cancer that affects white blood cells (lymphocytes) and thus we think that the blood is an ideal sample type for measuring proteins in studies of risk of haematological

malignancies such as NHL and hence the large number of hits is what might be anticipated. The blue and grey points are described in the legend for Figure 4, and we have now edited this for clarity. Please see below.

Figure caption page 12, Lines 495-496

Figure 4 – Mirror Manhattan plot for the association of genetically predicted protein concentrations and cancer risk using cis-pQTL and exome scores

This mirror Manhattan plot displays the results of each cis-pQTL (top) in the full exome-sequencing cohort within the UK Biobank across European samples for proteins passing correction for multiple testing in the observational results on cancer risk. The y-axis represents the $-\log_{10}$ p-values. The bottom of this plot contains the exome-wide score results for genetically predicted proteins. Markers colored in grey represent results that did not reach the conventional $p < 0.05$ significance threshold, while markers in blue represent conventionally significant results. If a cis-variant or an exome-wide score passed Bonferroni significance, those markers are colored by the cancer site of association. Purple markers represent non-Hodgkin lymphoma (NHL), light brown represents leukemia, green for liver cancer and pink for lung cancer. Red dash lines represent the threshold for Bonferroni multiple test comparison, with the yellow dash line representing the conventional significance threshold. Odds Ratios (OR) are the relative risk per standard deviation increase. Cis-variants were adjusted to be on the same scale.

20) Line 314: Why 533 out of 621?

Authors' Response

Similar to the sometimes-limited availability of cis-acting genetic variants from our answer above, there are some proteins where no genetic predictors have been identified in current studies and therefore cannot be investigated using exome-scores. This can be due to multiple reasons that include the relatively low abundance of some proteins in healthy individuals that can limit statistical power for genetic discovery. In our study, we were able to genetically proxy 533 of the 618 protein-cancer associations using exome-scores.

21. Misc comment: The figures are missing some panel labels. Mentions of figures in the text need to specify the panel.

Authors' Response

In response to your helpful comment about the figures, we have now updated Figure 1 and separated the volcano plots (now available in Extended Figures 7-31), and so no longer have figure panels.

Noninvasively collected biospecimen are useful sources for cancer risk assessment, disease stratification and generating risk-prediction models. This very well written and appropriately designed epidemiology association study provides timely and health relevant information about protein-based cancer detection. Methodology and interpretation of results is excellent.

Authors' Response

We thank the reviewer for their kind comments.

A few comments are:

1. Since most of the samples are from Caucasian population, how much this information would be applicable to other ethnic and racial groups?

Authors' Response

There are very few non-European ancestry high-throughput proteomic prospective studies and fewer still assess cancer risk and so it is difficult to assess the generalisability of our UK Biobank (primarily European ancestry) results to other populations. Two of the existing prospective cohort studies with available data for proteomics in non-European participants, the China Kadoorie Biobank (CKB) and the Atherosclerosis Risk in Communities (ARIC) Study, were designed to investigate cardiovascular disease. However, in one large cancer study designed to identify markers of imminent lung cancer diagnosis and led by one of the co-authors [2] , most proteins that were associated with lung cancer risk in European participants also were associated with risk among participants from the Singapore Chinese Health Study (SCHS). Nonetheless, two proteins strongly associated with risk of lung cancer among SCHS participants were not associated with risk in European participants in the same study. These limited data suggest that while there may be some generalisability to non-European populations, substantially more data are needed to understand which protein-cancer associations are ancestry-specific.

In the present study, our exome-based analyses were restricted to participants with European ancestry. Previous studies with available genetic and protein data among non-European participants found that greater than 90% of pQTL identified in European populations were present in non-European populations [4, 5]. However, one-third of pQTL identified in African-Ancestry and Han Chinese populations may not be present in European populations [4, 6]. These studies imply that findings for the proteins and cancer risk based on our exome-derived protein scores may provide some limited insights into aetiological proteins for cancer risk in other ancestries.

We have now expanded the discussion text on the generalisability of our protein findings and included some additional references in our revised text.

Discussion, p.11, Lines 440-442.

Proteomics holds significant promise for developing future cancer prevention initiatives that are needed to address the predicted increase in cancer burden among diverse populations, and so further studies into the proteomics of cancer risk including in non-White populations are necessary.⁴⁸ This is especially important as the limited observational and genetic evidence suggest that inherited determinants of proteins and the protein-cancer associations can vary between populations of different ancestry [2-4, 6, 7].

2. It would be useful to include information about any plan to validate some identified protein markers.

Authors' Response

We thank the Reviewer for their comments on validation and agree that this is an important next step. We have included results from genetic exome-based analyses as one form of validation study. As referenced by Reviewer #3, it is not currently possible to perform external validation for observational findings in UK Biobank using prospective data. An aim of this report is to generate discussion on best practices in the design and interpretation of such large-scale analyses, as well as to prime the generation of further prospective cohort data.

We plan to generate additional data on proteins for a variety of cancer endpoints within other mature prospective cohorts, such as the European Prospective Investigation into Cancer and Nutrition, and have now added information on this in our revised text.

Discussion, p.11, Line 445

There are several research priorities leading from our findings that are necessary to pursue to more fully understand the roles of proteins in cancer development and progression. Priorities are more large-scale prospective data from mature cohorts, such as in the European Investigation into Cancer and Nutrition (EPIC), to replicate our findings, and further complementary genetic studies, including Mendelian randomization analyses.

3. Any reason for identifying a small number of biomarkers for few cancer types?

Authors' Response

We also noted that for many common cancer types we identified few associations with circulating proteins, while for a small number of cancer types there were many associations. This latter group of cancers were those relating to the blood or in tissues related to maintaining blood composition, i.e. liver, renal, and lung cancer. It may therefore be easier to ascertain signals from these cancers using blood-measured protein concentrations than for the cancers at other sites.

Further, the proteins available for analysis in our study represent a sizable but modest subset of the total human proteome (~18k proteins) and are primarily those selected by Olink, as they can be quantified in the circulating plasma. They are necessarily therefore biased towards those with higher abundance in the plasma and, in their current form, these technologies cannot be guaranteed to measure proteins with important aetiological roles that are specific to tissues other than the blood. Therefore, the low number of protein-cancer associations for some cancer types may also reflect the fact that not all aetiologically relevant proteins were included in these Olink panels. We have now further discussed this in our revised text.

Discussion, p.9, Lines 351-353.

*We found that protein-cancer associations were most prevalent for cancers related to the blood or in tissues with a role in the maintenance of blood composition or with a high throughput of blood, such as the liver, kidneys, and lungs. Further, the smaller number of protein associations for cancers with higher incidence in this study but whose organs are not directly involved in blood composition (such as breast and prostate) may indicate a more localized effect and highlight the limitation of only measuring blood protein levels when investigating diseases in other tissues. **We note that given current multiplex technologies measure a sizable but limited subset of the total human proteome, it is possible that because of this ascertainment bias in currently measurable proteins may not include all those with important aetiological relevance for cancer risk.** When, in the future, stage and histological grading information becomes available for cancers within the UK Biobank or other cohorts, it may be possible to identify proteins associated with disease that has progressed beyond the primary organ that may lead to more easily measurable effects in the circulation.*

Reviewer #3

This paper is a study that confirmed the association between plasma proteins and cancer risk in 44,645 people from the UK biobank. The validity of the protein marker is partially proven using a method to confirm the protein marker through cancer risk analysis of QTL/exGS among all biobank participants. There are currently few such studies, so it is not possible to confirm their external validity. In this study, pathogenic protein markers were selected using exome score and expression information, and we believe this method is the most appropriate method at present. Future research should be conducted to confirm whether the protein markers identified in this study are the same in other population studies, that is, to confirm external validity. In this study, eligible criteria of subjects were appropriate and the analysis method was well described. The analysis of markers was also well described, and it was also appropriate to set a period of > 7 years (from blood collection to cancer diagnosis) to see the effect of protein markers on cancer development. Thus the proteins ascertained in this study are thought to be proper markers in prediction of future cancer development.

Authors' Response

We thank the reviewer for recognising the contribution of this work and for their kind words regarding the methodological techniques we employed to provide triangulated validity our findings by integrating data on the circulating protein from different data sources with orthogonal biases [8]. We also thank the Reviewer for highlighting that there are no other large and mature prospective epidemiological cohorts with high throughput proteomics data in relation to cancer incidence, but we anticipate such studies will be possible in the future.

1. The exome genetic score and protein profile you analysed were not presented in the supplementary. Please include the UK biobank website where such profiles are presented in the methods of this paper. Please include the UK biobank website where such profiles are presented in the methods of this paper for reference by other researchers.

Authors' Response

We thank the reviewer for this suggestion. The protein concentrations are available for download via the remote access platform hosted by DNA nexus and for download via the central UK Biobank website where researchers add these data through the baskets. With respect to the exome-based protein scores, the weights for these protein profiles and the code to generate them can be found on our GitHub and we now reference in this under the Data access section in our revised text. Additionally, we look forward to returning these individual-level protein profiles to the UK Biobank so that they may facilitate future research within the UK Biobank resource.

Data access, p.11, Lines 463-466.

Data access:

This research has been conducted using the UK Biobank Resource under Application Number 67506. All bona fide researchers can apply to use the UK Biobank resource for health related research that is in the public interest (<https://www.ukbiobank.ac.uk/register-apply/>). We wish to express our gratitude to the participants and those involved in building the resource. The code and weight for the protein profiles generated in this study can be found:https://github.com/GenomicEPIOX/paper_1463_proteins_19_cancers_UKBB/tree/main. Exome-based protein scores generated as part of this study will be returned to the UK Biobank and made available to researchers in due course.

References

1. Álvez, M.B., F. Edfors, K. von Feilitzen, M. Zwahlen, A. Mardinoglu, P.H. Edqvist, T. Sjöblom, et al., *Next generation pan-cancer blood proteome profiling using proximity extension assay*. Nat Commun, 2023. **14**(1): p. 4308.
2. Lung Cancer Cohort Consortium (LC3), *The blood proteome of imminent lung cancer diagnosis*. Nat Commun, 2023. **14**(1): p. 3042.
3. Sun, B.B., J. Chiou, M. Traylor, C. Benner, Y.H. Hsu, T.G. Richardson, P. Surendran, et al., *Plasma proteomic associations with genetics and health in the UK Biobank*. Nature, 2023. **622**(7982): p. 329-338.
4. Zhang, J., D. Dutta, A. Köttgen, A. Tin, P. Schlosser, M.E. Grams, B. Harvey, et al., *Plasma proteome analyses in individuals of European and African ancestry identify cis-pQTLs and models for proteome-wide association studies*. Nat Genet, 2022. **54**(5): p. 593-602.
5. Thareja, G., A. Belkadi, M. Arnold, O.M.E. Albagha, J. Graumann, F. Schmidt, H. Grallert, et al., *Differences and commonalities in the genetic architecture of protein quantitative trait loci in European and Arab populations*. Hum Mol Genet, 2023. **32**(6): p. 907-916.
6. Saredo Said, Alfred Pozarickij, Kuang Lin, Sam Morris, Christiana Kartsonaki, Niel Wright, Hannah Fry, et al., *Ancestry diversity in the genetic determinants of the human plasma proteome and associated new drug targets*. MedRxiv, 2023.
7. Dhindsa, R.S., O.S. Burren, B.B. Sun, B.P. Prins, D. Matelska, E. Wheeler, J. Mitchell, et al., *Rare variant associations with plasma protein levels in the UK Biobank*. Nature, 2023. **622**(7982): p. 339-347.
8. Munafò, M.R., J.P.T. Higgins, and G.D. Smith, *Triangulating Evidence through the Inclusion of Genetically Informed Designs*. Cold Spring Harb Perspect Med, 2021. **11**(8).

REVIEWERS' COMMENTS

Reviewer #1 (Remarks to the Author):

Authors have done a good job addressing reviewer concerns. Only one minor point below.

Line 136 'Exome-sequencing data preparation and quality control procedures in the UK Biobank have been previously described.' If a peer reviewed manuscript is not available to cite and authors wish to cite the website, then authors should specify the date the website was accessed. The site says at the bottom it was "Updated 5 months ago", so content accessed by authors may not be the same as content accessed by future readers.

Reviewer #1 (Remarks on code availability):

I do not routinely use Jupyter/iPython notebooks so I am unable to comment on the quality or reproducibility of the code.

Reviewer #2 (Remarks to the Author):

My concerns have been resolved.

Reviewer #3 (Remarks to the Author):

The authors responded appropriately to comments from reviewer and attached the proper corrected version.

Reviewer #3 (Remarks on code availability):

The site has the data dictionary, database, and analysis code.

Response to reviewers

Re: Identifying proteomic risk factors for cancer using prospective and exome analyses: 1,463 circulating proteins and risk of 19 cancers in the UK Biobank

We thank the reviewers for their comments, these have greatly improved our manuscript. We have addressed the remaining reviewers' comment below. We present reviewers' comments below, followed by our responses (indented). Revised manuscript text appears as highlighted text.

Reviewer reports

Reviewer #1

Authors have done a good job addressing reviewer concerns. Only one minor point below.

1. Line 136 'Exome-sequencing data preparation and quality control procedures in the UK Biobank have been previously described.' If a peer reviewed manuscript is not available to cite and authors wish to cite the website, then authors should specify the date the website was accessed. The site says at the bottom it was "Updated 5 months ago", so content accessed by authors may not be the same as content accessed by future readers.

Authors' Response:

We thank the reviewer for their comment and agree that this is important information. We have now added the date the website was accessed in the reference list.

20. Protocol for Processing UKB Whole Exome Sequencing Data Sets.
<https://dnanexus.gitbook.io/uk-biobank-rap/science-corner/whole-exome-sequencing-oqfe-protocol/protocol-for-processing-ukb-whole-exome-sequencing-data-sets>.
Website accessed on July 25 20, 2023.

Reviewer #1 (Remarks on code availability):

I do not routinely use Jupyter/iPython notebooks so I am unable to comment on the quality or reproducibility of the code.

Reviewer #2

My concerns have been resolved.

Reviewer #3

The authors responded appropriately to comments from reviewer and attached the proper corrected version.

Reviewer #3 (Remarks on code availability):

The site has the data dictionary, database, and analysis code.